# Maqui Berry and Ginseng Extracts Reduce Cigarette Smoke-Induced Cell Injury in a 3D Bone Co-Culture Model

**DOI:** 10.3390/antiox11122460

**Published:** 2022-12-14

**Authors:** Huizhi Guo, Weidong Weng, Shuncong Zhang, Helen Rinderknecht, Bianca Braun, Regina Breinbauer, Purva Gupta, Ashok Kumar, Sabrina Ehnert, Tina Histing, Andreas K. Nussler, Romina H. Aspera-Werz

**Affiliations:** 1BG Trauma Center Tübingen, Siegfried Weller Research Institute, Department of Trauma and Reconstructive Surgery, University of Tübingen, Schnarrenbergstr. 95, D-72076 Tübingen, Germany; 2Spine Surgery Department, The First Affiliated Hospital of Guangzhou University of Chinese Medicine, 12 Airport Road, Guangzhou 510405, China; 3Biomaterial and Tissue Engineering Group, Department of Biological Sciences and Bioengineering, Indian Institute of Technology Kanpur, Kanpur 208016, India; 4Centre for Nanosciences, Indian Institute of Technology Kanpur, Kanpur 208016, India; 5Centre for Environmental Sciences and Engineering, Indian Institute of Technology Kanpur, Kanpur 208016, India

**Keywords:** maqui berry, ginseng, osteoblast, osteoclast, co-culture system, cigarette smoke

## Abstract

Cigarette smoking-induced oxidative stress has harmful effects on bone metabolism. Maqui berry extract (MBE) and ginseng extract (GE) are two naturally occurring antioxidants that have been shown to reduce oxidative stress. By using an osteoblast and osteoclast three-dimensional co-culture system, we investigated the effects of MBE and GE on bone cells exposed to cigarette smoke extract (CSE). The cell viability and function of the co-culture system were measured on day 14. Markers of bone cell differentiation and oxidative stress were evaluated at gene and protein levels on day 7. The results showed that exposure to CSE induced osteoporotic-like alterations in the co-culture system, while 1.5 µg/mL MBE and 50 µg/mL GE improved CSE-impaired osteoblast function and decreased CSE-induced osteoclast function. The molecular mechanism of MBE and GE in preventing CSE-induced bone cell damage is linked with the inhibition of the NF-κB signaling pathway and the activation of the Nrf2 signaling pathway. Therefore, MBE and GE can reduce CSE-induced detrimental effects on bone cells and, thus, prevent smoking-induced alterations in bone cell homeostasis. These two antioxidants are thus suitable supplements to support bone regeneration in smokers.

## 1. Introduction

Due to constant contact with the environment, free radicals are generated in the human body through factors such as respiration (oxidative reaction), external pollution, and radiation exposure [1]. Numerous human diseases, including cancer, metabolic bone disease, and aging, are closely associated with excessive production of free radicals [2,3]. Free radical-induced oxidative stress damage can be attenuated effectively by antioxidants [4]. Therefore, natural antioxidant intake is popular worldwide for the promotion of healthy life. However, there is a lack of information about the effects of several plant-based extracts with antioxidant properties on bone metabolism. To evaluate the influence of these promising extracts on bone homeostasis, we need an effective and stable model representing bone metabolism in vitro. Bone metabolism depends on the mutual regulation between osteoblasts and osteoclasts. The imbalance between bone formation and bone resorption impairs bone homeostasis and, consequently, decreases bone mineral content [5]. In a co-culture system of osteoblasts and osteoclasts, osteoclast differentiation was influenced by cytokines released by osteoblasts, such as receptor activator of NF-κB ligand (RANKL) and osteoprotegerin (OPG), while osteoblasts were also regulated by cytokines secreted by osteoclasts, such as platelet-derived growth factor (PDGF) [6]. In our previous study, we found that both platelet-rich plasma (PRP) and gelatin (GEL) scaffolds were good carriers for the mono-culture of osteoblasts and osteoclast cultivation and differentiation [7]. Nevertheless, Wu et al. [8] showed that a PRP-coated scaffold was able to reduce oxidative stress in a rabbit model. As it is preferable to use a culture system without antioxidant properties to investigate the antioxidant properties of natural plant-based extracts, GEL scaffolds were chosen as a platform for our bone cell co-culture system.

Smoke from cigarettes contains more than 6000 chemicals, over 150 of which are toxic compounds [9]. It is known that nicotine and carbon monoxide from smoking can lead to cellular damage and multiple organ injury via induced oxidative stress [10]. Since 22.3% of the population worldwide are smokers [11], cigarette smoke-induced oxidative stress has received much attention. Many studies have demonstrated that smoking-induced oxidative stress is also harmful to bone metabolism [12,13,14]. Smoking can decrease osteoblast function and increase osteoclast function through the upregulation of oxidative stress levels [14]. Several studies have confirmed that smoking decreased the proliferation and differentiation of osteoblasts in a concentration-dependent manner [14,15]. In addition, smoking-induced oxidative stress also results in accelerated, more efficient osteoclastogenesis, thus increasing the resorption of bone [15]. Due to the disruption of bone homeostasis, cigarette smoking negatively affects bone fracture healing. Clinical studies have shown that patients who smoke have longer hospital stays, more complications, and increased risk of delayed or nonunion bone fracture healing [12,16]. Therefore, it is mandatory to find treatments or prevention strategies to counteract cigarette smoking-impaired bone metabolism.

Numerous studies have revealed that antioxidants can exert protective effects on bone homeostasis by inhibiting oxidative stress and, therefore, could be a beneficial supplement to support bone regeneration in smokers [17,18,19]. Maqui berry, a fruit native to South America, has been reported to have high antioxidant activity due to its rich content of anthocyanins and tea polyphenols [20]. Maqui berry has many pharmacological properties, including antioxidant [21], anti-aging [22], hypoglycemic [23], and anti-inflammatory properties [24]. As a standard extract product, Delphinol^®^ maqui berry extract (MBE) contains 25% delphinidin (one of the anthocyanins) and is the richest natural source of delphinidins known to date [18]. Our previous study showed that MBE enhanced primary human osteoblast functions in CSE-induced oxidative stress damage via activation of the antioxidant defense system [17]. However, the effects of MBE on the interaction between osteoblasts and osteoclasts and their influence on osteoclast functions have not been investigated until now.

Ginseng, an important herbal drug that is primarily grown in Korea and northeast China, has traditionally been administered to treat various diseases for thousands of years [25]. Ginseng extract (GE) is a powdered extract that is produced from ginseng roots by an aqueous-ethanolic solvent. Approximately 38 types of ginsenosides and various polysaccharides have been found in GE [25,26]. These active ingredients have extensive bioactivities, including antifatigue, antioxidant, anti-inflammatory, and antitumor activities [27,28,29]. Ginsenosides effectively inhibit mitochondrial damage-mediated reactive oxygen species (ROS) production by activating the nuclear factor erythroid 2-related factor 2 (Nrf2) signaling pathway [30]. Kim and colleagues also found that ginseng neutral polysaccharide could decrease the ROS concentration in vivo [28]. Furthermore, some research has revealed that GE can improve bone quality by inhibiting osteoclast-induced bone resorption [31]. Therefore, the use of GE could be a potential functional prevention strategy to promote bone health in smokers with high oxidative stress.

Nrf2 signaling is a crucial pathway that is involved in modulating oxidative stress [32]. Usually, Nrf2 level and transcriptional activity are relatively low because of proteasomal degradation due to the interaction with the Kelch-like ECH-associated protein 1 (Keap1) [33]. Under stress conditions, Nrf2 is liberated from the Keap1-Nrf2 complex and translocated to the nucleus, where it binds the antioxidant response element (ARE) sequences in the promoter regions on antioxidant genes [34]. Antioxidants, such as N-acetyl cysteine (NAC) and MBE, have been shown to protect bone-like cells against ROS and related cell injury by activating Nrf2 signaling [13,17].

Nuclear factor kappa-B (NF-κB) is a member of the pleiotropic transcriptional factor family and comprises NF-κB1 (P50), NF-κB2 (P52), RelA (P65), RelB, and c-Rel [35]. NF-κB plays an important role in bone metabolism, inflammation, and the immune response, while the action of NF-κB is regulated by the inhibitor of kappa B (IκB) [36]. The regulatory effect of the NF-κB signaling pathway on osteogenesis- and osteoclastogenesis-related gene expression is critical to maintain bone cell homeostasis [37]. In osteoclastogenesis, NF-κB pathways were the first known signaling pathways triggered by RANKL stimulation regulating osteoclast formation and function [38]. Animal experiments revealed that NF-κB1/NF-κB2 double knockout mice exhibited severe osteopetrosis due to the complete lack of mature osteoclasts [39], suggesting that activation of NF-κB is essential for osteoclast differentiation. In addition, silencing of IκB kinase (which enables NF-κB nuclear translocation) in mouse osteoblasts increased the bone mass with no effect on osteoclast number [40]. Therefore, inhibition of NF-κB is considered to both suppress bone resorption and promote bone formation.

Accordingly, by establishing an effective osteoblast and osteoclast 3D co-culture system in scaffolds, this research aimed to investigate the roles of MBE and GE in preventing smoking-induced bone cell damage and elucidate the associated molecular mechanism.

## 2. Materials and Methods

### 2.1. Chemical Reagents, Cell Culture Medium

All the chemicals were purchased either from Carl Roth (Karlsruhe, Germany) or Sigma-Aldrich (St. Louis, MO, USA). Cell culture mediums and other supplements were obtained from Sigma-Aldrich or Gibco (Thermo Fisher Scientific, Waltham, MA, USA).

### 2.2. The Preparation of GEL Scaffolds

The fabrication procedure has already been described in our previous research [7]. Briefly, for construction of the GEL scaffold, gelatin from cold-water fish skin (G7041, Sigma, St. Louis, MO, USA) and hydroxyapatite (21223, Sigma, St. Louis, MO, USA) were thoroughly mixed in a 50 mL falcon tube to achieve final concentrations of 4.8% and 10%, respectively. Following the addition of 1% glutaraldehyde (3778.1 Carl Roth, Karlsruhe, Germany), the mixture solution was immediately transferred into polystyrene casting molds and frozen overnight at −18 °C. After incubation at −80 °C for 1 h, the formed matrix was cut into a cylindrical shape (height: 3 mm; diameter: 6 mm) with the help of a razor blade.

To achieve optimal sterilization, GEL scaffolds were incubated with 70% ethanol overnight on a rotating shaker. Following washing with phosphate buffer saline (PBS; D8537, Sigma, St. Louis, MO, USA) three times, the scaffolds were incubated in culture medium as a sterile control.

### 2.3. Cell Lines

SCP-1 cells (kindly provided by Dr. Matthias Schieker [41]), a single-cell-derived human bone marrow mesenchymal stem cell line (hMSC), were used as osteogenic precursor cells [7]. THP-1 cells, a human acute monocytic leukemia cell line, were used as osteoclast precursor cells [42]. SCP-1 cells and THP-1 cells were cultured with Minimum Essential Medium Eagle Alpha (MEM α; 22561-054, Thermo Fisher Scientific, Waltham, MA, USA) or RPMI 1640 medium containing 5% fetal bovine serum (FBS; 41G7141K, Thermo Fisher Scientific, Waltham, MA, USA) in a 5% CO_2_, 100% humidity incubator at 37 °C, respectively. The medium was changed twice a week. Cells were sub-cultured when they reached confluency. Cell lines were used below passage 20.

### 2.4. The Construction of the 3D Bone Cell Co-Culture System

After removing the medium from pre-conditioned scaffolds, 15 μL of THP-1 cell suspension (8 × 10^4^ cells/scaffold) was dripped into the scaffolds. Following a 4 h incubation at 37 °C, 500 µL of cell culture medium supplemented with 200 nM phorbol 12-myristate 13-acetate (AB120297, Abcam, Cambridge, UK) was gently added. After 24 h, 15 μL of SCP-1 cell suspension (1 × 10^4^ cells/scaffold) was dripped into the same scaffold. After 4 h incubation at 37 °C, 500 µL of osteogenic medium (50:50 mix of RPMI to MEM α, 2% FBS, 200 μM L-ascorbic acid 2-phosphate (A8960-5G, Sigma), 5 mM β-glycerolphosphate (G9422-10, Sigma, St. Louis, MO, USA), 25 mM HEPES (HN78.2, Carl Roth, Karlsruhe, Germany), 1.5 mM CaCl_2_ (CN93.1, Carl Roth, Karlsruhe, Germany), and 5 μM cholecalciferol (95230, Sigma, St. Louis, MO, USA) was added [42]. The osteogenic medium was renewed twice a week/every two to three days.

### 2.5. The Preparation of Cigarette Smoke Extract (CSE) and Antioxidants

CSE was always prepared freshly on the day of the medium change. Through a peristaltic pump device, smoke from one commercial cigarette (Marlboro, New York, NY, USA) was continuously drawn and dissolved into a 25 mL plain culture medium (without FBS and osteogenic induction factors) [43]. The optical density of the CSE was normalized at 320 nm (OD_320_), with an OD_320_ of 0.65~0.7 considered 100% CSE [44]. After sterile filtration with a 0.22 μm pore filter, 100% CSE was diluted with osteogenic medium to a final concentration of 5%, which was equivalent to smoking 10 cigarettes per day [45]. Maqui berry extract (MBE), ginseng extract (GE), and N-acetylcysteine (NAC, 616-91-1, Carl Roth) were dissolved in plain culture media and sterilized with 0.22 μM pore filters. The stock concentrations of MBE, GE, and NAC were 1.5 mg/mL, 3 mg/mL, and 70 mM, respectively. At the time of use, antioxidant stock solutions were diluted with culture medium to the required concentrations. MBE and GE were produced according to the requirements of the Pharmacopoeia European monograph by Anklam Extrakt GmbH, Anklam, Germany.

### 2.6. Resazurin Conversion Assay

The resazurin conversion assay was used to measure the mitochondrial activity, which can reflect the number of viable cells on the scaffolds [13]. Before measurement, the scaffolds were washed with PBS and transferred to a new 48-well plate. Then, 500 μL of 0.0025% resazurin solution (199303-1G; Sigma) was added per well to cover the scaffolds. After incubation for 2 h at 37 °C, 100 μL/well solution was transferred into a 96-well plate and the fluorescence at 544 nm excitation and 590 nm emission wavelengths was measured with an Omega Plate Reader (BMG Labtech, Ortenberg, GER). The values of resazurin conversion from scaffolds without cells were subtracted as background.

### 2.7. Total DNA Isolation and Quantification

DNA was isolated using the alkaline lysis method. After being washed with PBS, 250 μL/well of 50 mM hot (98 °C) NaOH was added and incubated for 15 min. Then, scaffolds with NaOH solution were frozen at −80 °C overnight. The scaffolds were heated at 60  °C for 20 min until fully melted. An equal volume of 100 mM Tris buffer (pH = 8.0) was used to adjust pH values. A total of 100 μL of each sample was transferred into a V-bottom 96-well plate to remove impurities by centrifuging (1000× *g*, 10 min). DNA concentration was measured photometrically with an LVIS plate and Omega Plate Reader (wavelength λ = 230 nm, λ = 260 nm, and λ = 280 nm; 25 flashes) [42].

### 2.8. Alkaline Phosphatase (AP) Activity

AP activity was used as an early osteogenic marker [13,43]. After being washed three times with PBS, scaffolds were immersed in 500 μL of AP reaction buffer (0.2% *w*/*v* 4-nitrophenyl-phosphate (N7660, Sigma, St. Louis, MO, USA), 50 mM glycine (3908.2, Carl Roth, Karlsruhe, Germany), 1 mM MgCl_2_ (3908.2, Carl Roth, Karlsruhe, Germany), and 100 mM TRIS; pH 10.5) to react for 2 h. The conversion of 4-nitrophenyl-phosphate to 4-nitrophenol was quantified photometrically (λ = 405 nm; Omega Plate Reader, BMG Labtech). Experimental values were corrected to background control (scaffold without cells). After that, data were normalized to viable cell numbers by resazurin conversion as previously reported [46].

### 2.9. Tartrate-Resistant Acid Phosphatase (TRAP) 5b Activity

TRAP 5b mainly arises from osteoclasts and is a specific marker for osteoclast activity [14]. A total of 30 μL of supernatant was mixed with 90 μL of TRAP 5b activity assay solution (0.2% *w*/*v* 4-nitrophenyl phosphate, 100 mM sodium acetate (X891.2, Carl Roth, Karlsruhe, Germany), and 50 mM sodium tartrate (4165.1, Carl Roth, Karlsruhe, Germany) in demineralized water; pH 5.5) to react for 6 h (37 °C). After adding 90 μL/well of 1 M NaOH (T135.1, Carl Roth, Karlsruhe, Germany) to suppress the reaction, the reaction product (4-nitrophenol) was measured with a photometer (λ = 405 nm; Omega Plate Reader, BMG Labtech). Experimental values were corrected to background control (scaffold without cells). After that, data were normalized to viable cell numbers by resazurin conversion as previously described [46].

### 2.10. Carbonic Anhydrase II (CAII) Activity

CAII activity was used as an early indicator of osteoclast differentiation [46]. Scaffolds were incubated with 500 μL of substrate solution (12.5 mM Tris, 75 mM NaCl (S7653, Carl Roth, Karlsruhe, Germany), pH 7.5; 200 mM 4-nitrophenyl acetate) for 15 min at 37 °C. Then, 100 μL solution (with formed pNP) was observed photometrically at λ = 405 nm with the Omega Plate Reader. Experimental values were corrected to background control (scaffold without cells). After that, data were normalized to viable cell numbers by resazurin conversion [46].

### 2.11. RT-PCR Analysis

Gene expression levels were determined by RT-PCR. Self-made TriFast (38% *v*/*v* phenol, 0.4 mM ammonium thiocyanate (221988, Sigma, St. Louis, MO, USA) 0.8 mM guanidine thiocyanate, 0.68 mM glycerol (55290, Otto Fischer GmbH, Waldkirch, Germany), and 0.1 M sodium acetate solution) was used to extract RNA from the scaffolds. To remove impurities, the solution was centrifuged three times at 14,000× *g* for 10 min before chloroform/phenol extraction. The RNA concentration was measured with the Omega Plate Reader. The cDNA synthesis kit (Thermo Fisher Scientific, Waltham, MA, USA) was used to synthesize cDNA. Then, PCR amplification was performed according to the manufacture instructions for Biozym Red HS Taq Master Mix (Vienna, Austria). Primer sequences and PCR conditions for target genes are shown in Table 1. 18S rRNA served as a housekeeping gene. A 1.8% (*w*/*v*) agarose gel with ethidium bromide was used to visualize the PCR products. The electrophoresis was carried out at 90 V for 60 min to separate the PCR products. Finally, the intensity of the bands was quantitatively analyzed by ImageJ (NIH, Bethesda, MD, USA) [47].

### 2.12. Dot Blot Analysis

Secreted proteins in culture supernatant were detected by dot blot, as previously described [7]. Using a 96-well dot blotter (Carl Roth, Karlsruhe, Germany), 60 μL of supernatant was transferred onto a wet nitrocellulose membrane with a vacuum pump. After being blocked with 5% bovine serum albumin in Tris-buffered saline/Tween 20 (TBS-T (9127.1, Carl Roth, Karlsruhe, Germany); 10 mM Tris-HCl at pH 7.6, 0.15 mM NaCl, and 0.1% Tween-20) for 1 h, membranes were incubated with primary antibodies at 4 °C overnight. The antibodies used are summarized in Table 2. After washing with TBS-T, the membranes were incubated with secondary antibody for 2 h. The signals were detected by chemiluminescence (100 nM TRIS, 250 mM Luminol (4203.1, Carl Roth, Karlsruhe, Germany), 90 mM *p*-coumaric acid (9908.1, Carl Roth, Karlsruhe, Germany), 30% *v*/*v* H_2_O_2_ (CP26.5, Carl Roth, Karlsruhe, Germany)) and quantified by ImageJ (NIH, Bethesda, MD, USA).

### 2.13. Western Blot

Protein expression levels were detected by Western blot (WB). Cells were lysed in 1× RIPA buffer containing protease/phosphatase inhibitors (1 μg/mL Pepstatin, 5 μg/mL Leupeptin, 1 mM phenylmethylsulfonyl fluoride, 5 mM NaF, and 1 mM Na_3_VO_4_). Total protein lysates were then centrifuged at 14,000× *g* for 10 min at 4 °C to remove cell debris. The protein concentrations were determined with the micro-Lowry method [15]. As previously described [47], 25 μg of total protein was separated by 12% acrylamide gels (100 V, 180 min) and then transferred onto a nitrocellulose blotting membrane (100 mA, 180 min). Ponceaus staining was used to visually ensure adequate protein transfer. After being blocked with 5% BSA for 1 h, primary and secondary antibody incubations were performed as previously described for the dot blot (Section 2.12). The antibodies used are summarized in Table 3. The chemiluminescent signals in bands were visualized with a charge-coupled device camera and quantified in ImageJ (NIH, Bethesda, MD, USA).

### 2.14. Immunofluorescence Staining

Immunofluorescence was used to visualize the translocation of NF-κB protein in the nucleus. After 7 days co-culture in a 96-well plate, SCP-1 and THP-1 cells were fixated with 4% formaldehyde solution (4979.1, Carl Roth, Karlsruhe, Germany) for 15 min. Then, 1% Triton-X-100 (3051.2, Carl Roth, Karlsruhe, Germany) solution was added for incubation for 5 min. The cells were blocked with 5% BSA for 1 h and incubated with an anti-NF-κB antibody (1:1000 in PBS; sc-109, Santa Cruz, Heidelberg, Germany) overnight at 4 °C. The following day, cells were washed with PBS and incubated with Alexa-488-labeled secondary antibody (1:2000 in PBS, Invitrogen, Karlsruhe, Germany) and Hoechst 33342 (1:1000 in PBS; 14533, Sigma, St. Louis, MO, USA) for 2 h. After washing with PBS, actin cytoskeleton was stained with phalloidin-tetramethylrhodamine (1:2000 in PBS; P2141, St. Louis, MO, USA). Fluorescent images were taken and analyzed with a CELENA^®^ X High Content Imaging System (Logos Biosystems, Anyang, Korea).

### 2.15. Statistical Analysis

The data are presented as means ± the standard error of the mean (SEM). All the experiments were repeated at least three times with two or three technical replicates. Statistical analyses were performed using GraphPad Prism software (GraphPad Software 9.0, La Jolla, CA, USA). The data of the two groups were compared with the Mann–Whitney test. The data of multiple groups were compared with the non-parametric Kruskal–Wallis test, followed by Dunn’s multiple comparison test. A two-way ANOVA test followed by Turkey’s multiple comparisons was used when two independent variables were compared among groups. A *p* < 0.05 was considered statistically significant.

## 3. Results

### 3.1. Cytotoxicity Tests of MBE, GE, and NAC

To select nontoxic concentrations of antioxidants for the 3D bone co-culture system, cells were treated with increasing concentrations of MBE (1.5, 3, 6, 12, and 60 μg/mL), GE (25, 50, 75, 100, and 200 μg/mL), or NAC (1, 3.5, 7, 14, and 28 mM) for 14 days. The mitochondrial activity and total DNA content were used to indicate cell viability. MBE at a concentration ≥ 12 μg/mL showed cytotoxic effects on the SCP-1/THP-1 co-culture system (Appendix A). A high concentration of GE (≥200 μg/mL) decreased mitochondrial activity significantly and decreased total DNA in the co-culture system (Appendix A). As for NAC, cell viability showed a remarkable reduction when the concentration exceeded 7 mM (Appendix A). Previous studies also reported that 1.5 μg/mL MBE [17] and 50 μg/mL GE [48] did not have cytotoxic effects in vitro. Therefore, based on our results and the literature, 1.5 μg/mL MBE and 50 μg/mL GE were used for subsequent experiments. Since our previous study demonstrated that 3.5 mM NAC had a protective role in CSE-induced bone cell injury [13], we used NAC as a positive control.

### 3.2. MBE and GE Suppressed Osteoclast Function in Bone Cells Exposed to CSE

To assess the effects of antioxidants on bone cells exposed to CSE, cell viability and functionality were measured after 14 days. The 3D co-culture system was exposed to 5% CSE in combination with 1.5 μg/mL MBE, 50 μg/mL GE, or 3.5 mM NAC. On day 14, co-cultures exposed to 5% CSE showed a significant reduction in mitochondrial activity (*p* ˂ 0.0001; Figure 1a) and total DNA content (*p* ˂ 0.0001; Figure 1b) relative to untreated cells. Co-incubation with MBE or GE slightly enhanced the viability (mitochondrial activity and total DNA content) of CSE-treated co-cultures on day 14. As expected, NAC caused a significant increase in mitochondrial activity (*p* ˂ 0.05; Figure 1a) and total DNA content (*p* ˂ 0.05; Figure 1b) in co-cultures treated with 5% CSE.

As well as cell viability, osteoclast function (CAII and TRAP activity) was measured in the co-culture systems. CAII and TRAP activities are early and late indicators of osteoclast differentiation, respectively [46]. Our CAII and TRAP activity results (Figure 1c,d) showed that osteoclast function was markedly increased in co-cultures exposed to 5% CSE, contributing to the development of an osteoporotic bone environment in vitro. Most interestingly, co-incubation with MBE or GE significantly reduced osteoclast function almost to control levels in bone cells exposed to CSE after 14 days (Figure 1c,d).

### 3.3. MBE and GE Enhanced Osteogenic Differentiation and Inhibited Osteoclastic Differentiation in Bone Cells Exposed to CSE

Expression levels of osteoblastic genes (*Collagen 1* and *Runx2*) were investigated by RT-PCR. *Collagen 1* provides structural support for bone cells but also forms a network to promote bone mineralization [49]. *Runx2* is a master transcription factor of osteoblasts that can promote osteogenic differentiation of MSCs [50]. After 7 days, the expression of *Collagen 1* and *Runx2* decreased by 54.2% and 35.6% in cells exposed to 5% CSE, although those results were not statistically significant (Figure 2a–c). Following MBE, GE, or NAC treatment, *Collagen 1* and *Runx2* expression were significantly upregulated in bone cells exposed to CSE, reaching control levels (Figure 2a–c).

The osteoclastic genes (*NFATc1*, *TRAP 5b*, *MMP-9*) were also analyzed after 7 days of co-incubation with CSE and antioxidants. *NFATc1* is an essential transcription factor in the terminal differentiation of osteoclasts [36]. *TRAP 5b* and *MMP9* are required for the proteolytic activity of osteoclasts needed to resorb bone matrix [51]. These two genes are characteristic markers of osteoclast differentiation [51]. After 7 days of CSE exposure, the gene expression of *NFATc1*, *TRAP 5b*, and *MMP-9* significantly increased (*p* ˂ 0.01; Figure 2a,d–f). MBE or GE treatment showed a tendency to downregulate the expression of these osteoclastic marker genes in co-cultures exposed to CSE (Figure 2a,d,e). Similarly, osteoclastic marker gene expression was also downregulated by NAC treatment in bone cells exposed to CSE (Figure 2a,f).

### 3.4. MBE and GE Enhanced Osteoblast Function in Bone Cells Exposed to CSE

To evaluate the influence of MBE and GE on CSE-impaired osteogenesis, secreted AP, OCN, and PINP in the supernatant were detected by dot blot after culture of the system for 14 days. AP is produced in the early stages of osteogenic differentiation and can reflect osteoblastic activity [52]. OCN is a non-collagenous protein that is formed during the period of bone matrix mineralization [53]. Type I collagen is the major collagenous protein in the bone matrix [46]. Following the synthesis of new type I collagen, PINP is cleaved from type I procollagen by proteases outside the osteoblast. Thus, PINP could reflect the degree of bone mineralization [54]. After adding 5% CSE to the culture media, a significant decrease in AP levels (*p* ˂ 0.05; Figure 3a,b), a tendency towards lower OCN levels (Figure 3a,c), and a significant decrease in PINP levels (*p* ˂ 0.01; Figure 3a,d) were observed compared to untreated cells. MBE and GE treatment showed a tendency to upregulate these secreted osteoblast makers in co-cultures exposed to CSE (Figure 3a–d) after 14 days. Similar to MBE and GE, NAC treatment enhanced osteoblast markers in bone cells exposed to CSE (Figure 3a–d).

### 3.5. MBE and GE Reduced CSE-Induced Cell Injury by Downregulation of sRANKL: OPG Ratio and NF-κB Signaling Pathways

Both sRANKL and OPG are secreted by osteoblasts. sRANKL is a critical cytokine for osteoclastogenesis and its deficiency prevents the formation of multinucleated osteoclasts from monocyte/macrophage lineage cells [55]. In contrast to sRANKL, OPG inhibits osteoclastic activity by decoying sRANKL [14]. Therefore, the ratio of sRANKL and OPG reflects the modulatory ability of osteoblasts towards osteoclasts. The supernatant of the co-culture system treated with CSE and antioxidants was collected on day 14 to measure the secreted protein levels. The results showed that the amount of sRANKL secreted rose in the CSE group and fell in the MBE, GE, and NAC groups (Figure 4a). As for the expression levels of secreted OPG, an increasing trend was also found in the MBE, GE, and NAC groups (Figure 4a). Compared with the CSE group, a dramatic reduction in the ratio of sRANKL and OPG occurred in the MBE, GE, and NAC groups (Figure 4b). This result indicates that MBE and GE prevent CSE-induced bone cell damage by downregulating the sRANKL/OPG ratio.

In osteoclastogenesis, NF-κB pathways were the first signaling pathways triggered by RANKL stimulation to be identified [38]. Therefore, we used WB and immunofluorescence to analyze NF-κB and p-ERK1/2 protein levels in the co-culture system on day 7. With the WB analysis, we could detect the total protein level of NF-κB in the co-culture system, while immunofluorescence staining showed the translocation of NF-κB in the nucleus. The WB results revealed that NF-κB total protein levels were similar in bone cells treated with antioxidants and/or CSE (Figure 4c,d). Interestingly, the protein expression of p-ERK1/2 was downregulated by MBE, GE, and NAC in comparison to CSE alone (Figure 4c,e). Immunofluorescence staining was further performed to investigate the translocation of NF-κB protein in the nucleus. Microscopy images showed that NF-κB protein was primarily expressed on osteoclasts rather than on osteoblasts (Appendix A). Therefore, the translocation level of NF-κB protein in the nucleus was only analyzed for osteoclast-like cells. Compared with the absence of fluorescence signal in the control group, high expression of NF-κB protein was present after CSE exposure (Figure 4f). The addition of MBE, GE, and NAC significantly decreased the nuclear fluorescent signal intensity (Figure 4g) compared to CSE. These results indicate that MBE and GE decrease CSE-induced osteoclast differentiation by regulating the sRANKL/OPG ratio and NF-κB signaling pathways.

### 3.6. MBE and GE Prevented CSE-Induced Cell Injury by Activating Nrf 2 Signaling Pathway

It is well-known that CSE induces oxidative stress in bone cells and that pre-, post-, and co-incubation with antioxidants reduce bone cell stress damage [12,13,16,17]. To further explore the molecular mechanism of MBE and GE in preventing bone cell damage associated with CSE, the protein levels of p-Nrf2 and SOD1 were analyzed by Western blot. The transcription factor Nrf-2 activates the antioxidant defense system by regulating the antioxidative enzyme SOD1. After 7 days’ differentiation of the system with antioxidants and (or) CSE, total protein was extracted from the co-culture systems. CSE exposure mildly increased the phosphorylation of Nrf2 and the level of SOD1 (Figure 5a). MBE and GE upregulated the expression of the activated Nrf2 and its target protein SOD1 by two- and threefold, respectively, compared to untreated systems (Figure 5b,c). As expected, NAC increased the protein levels of p-Nrf2 and SOD1 in response to CSE-associated oxidative stress. These results indicate that MBE and GE decrease CSE-induced bone cell damage by regulating Nrf2 signaling pathways. The molecular mechanism by which MBE and GE prevent CSE-induced bone cell damage is summarized in Figure 6.

## 4. Discussion

External stimuli (such as drugs, ultraviolet light, cigarette smoke, and ionizing radiation) and endogenous free radicals can directly or indirectly damage cellular components, such as proteins, lipids, and DNA [56]. To defend against these adverse effects, the body activates a complex oxidative stress response system to mitigate cellular damage. As a key transcription factor regulating antioxidant stress, Nrf2 plays an important role in inducing the cellular antioxidant response [57]. Keap1-Nrf2 is a classical regulatory pathway that controls oxidative stress levels. Keap1 is a substrate adaptor protein of the E3 ubiquitin ligase complex that can assemble with Cullin3 (Cul3) into a functional E3 ubiquitin ligase complex (Keap1-Cul3-E3) [58]. In homeostatic conditions, the Nrf-2-Keap1-Cul3-E3 complex is degraded by the proteasome system. When cells are exposed to oxidative stress, electrophilic stress induces a conformational change in the Keap1-Cul3-E3 ubiquitin ligase. This conformational change results in the inhibition of Nrf2 ubiquitylation and disruption of the Keap1-Nrf2 complex [59]. Following release, Nrf2 is activated by phosphorylation and binds with the antioxidant response element in the nucleus, leading to the activation of cytoprotective genes, such as superoxide dismutase (SOD), catalase (CAT), glutathione reductase (GR), and glutathione peroxidase (GPX) [60,61]. Our results showed that CSE with and without antioxidants activated the Nrf2-mediated antioxidant system. However, CSE showed negative effects on osteoblast survival and differentiation and, thus, the levels of Nrf2 and SOD1 after CSE exposure may not efficiently reduce the oxidative stress induced by CSE. Therefore, high oxidative stress-induced cytotoxicity inhibits osteogenic differentiation and mineralization. In contrast, MBE and GE promoted osteogenic differentiation and function by activating the Nrf2-mediated antioxidant system. We speculate that this led to a reduction in the expression of RANKL, an increase in the expression of OPG, and, thus, a drop in the ratio of RANKL/OPG.

In normal bone tissue, old bone is continuously destroyed by osteoclasts, while osteoblasts generate new bone to maintain bone homeostasis [62]. The interaction of osteoblasts and osteoclasts is involved in maintaining the integrity and mechanical strength of bone tissue [63]. Coupled bone cells contribute to osteogenesis and osteoclastogenesis through direct contact and indirectly through cytokine secretion [64]. First, gap junctions between osteoblasts and osteoclasts affect their growth, proliferation, differentiation, and migration. Second, osteoclasts can enhance osteoblast differentiation by releasing transforming growth factor-β (TGF-β), bone morphogenetic proteins (BMPs), and insulin-like growth factors (IGFs) [65]. Similarly, M-CSF and RANKL secreted by osteoblasts can activate osteoclast precursors and promote them to differentiate into mature osteoclasts [66]. By producing OPG and RANKL, osteoblasts regulate osteoclastogenesis and osteoclast function. OPG is an important receptor for RANKL, which inhibits the maturation of osteoclasts [67]. Overexpression of OPG inhibits the production of osteoclasts, leading to osteosclerosis, whereas deficiency of OPG enhances bone resorption and leads to osteoporosis [68]. Therefore, RANKL and OPG can affect bone resorption and bone density by regulating the generation and function of osteoclasts. The increased osteoblast function in MBE and GE groups could inhibit osteoclast function via the RANKL/OPG ratio. In addition, we noticed that MMP9 gene expression was lower in the NAC group than in the MBE and GE groups. As a strong oxidant scavenger, the inhibitory effect of NAC on MMP9 was also reported in myofibroblasts in previous research and attributed to docking to Zn^2+^ at the active site [69].

Since NF-κB is also a key molecule for osteoclastogenesis, we further analyzed the expression of the NF-κB pathway in the co-culture system. During osteoclast differentiation, NF-κB is the first signaling pathway activated by RANKL [70]. NF-κB knockout mice showed severe ossification due to the complete lack of osteoclasts, suggesting that the activation of NF-κB is essential for osteoclast differentiation [40]. In addition, NF-κB is an oxidative stress-sensitive transcription factor that is directly activated by ROS [71]. On the other hand, NF-κB also induces oxidative stress, indicating that there is a vicious cycle between NF-κB and oxidative stress [72]. Interestingly, we found that there were no differences in total NF-κB protein among the different groups. However, when we focused on the proteins in the nucleus, NF-κB protein expression was strongly decreased in the antioxidant group (MBE and GE) compared to the CSE group. These results show that although antioxidants do not influence the total protein NF-κB levels, antioxidants promote the nuclear translocation of the transcriptional factor. Overall, CSE indirectly regulates the expression of NF-κB through RANKL and directly activates the NF-κB signaling pathway through ROS, thereby promoting the expression of osteoclast-related genes, while MBE and GE can downregulate the NF-κB signaling pathway by reducing oxidative stress and enhancing osteoblast activity. Nagaoka and colleagues demonstrated that MBE also downregulated the nuclear translocation of NF-κB in MC3T3-E1 cells treated with lipopolysaccharide [73]. Additionally, several studies showed that GE also inhibited NF-κB nuclear translocation associated with inflammation in cancer cell lines [74,75].

Molecular interactions between the NF-κB and Nrf2 pathways have been reported [36]. Aside from its regulatory effect in the Nrf2 pathway, Keap1 has been confirmed to regulate the NF-κB signaling pathway via IKKβ [58]. IKKβ is an important component of the IKK kinase complex in the classical activation pathway of NF-κB. The enhanced activity of IKKβ could effectively upregulate the NF-κB signaling pathway [70]. Through binding with IKKβ, Keap1 induced the ubiquitination and degradation of IKKβ, thereby negatively regulating the activity of NF-κB. There is some evidence that Keap1 is involved in the functional interaction between the Nrf2 and NF-κB pathways [33,76]. Knockdown of the Keap1 gene in human umbilical vein endothelial cells not only upregulated the Nrf2-mediated antioxidant system but also suppressed the activation of the NF-κB signaling pathway [76]. Keap1 could be a key regulator of the interactions controlling cellular redox status and responses to stress. However, the interactions between Nrf2 and the NF-κB signaling pathway in the osteoblast and osteoclast co-culture system need to be further confirmed.

Delphinol^®^ is a standardized commercial extract from maqui berry. It contains a minimum of 25% delphinidins and 35% total anthocyanins. GE is extracted from ginseng roots, which consist of 10% total ginsenosides (natural glycosylated triterpenes also known as saponins). Those active components contained in MBE or GE have been described as possessing antioxidant properties and protective effects in several systems [46,47,48,49]. Therefore, we can hypothesize that the positive effects on bone cells were related to the active components of the extracts used in the study.

Several in vitro and in vivo studies have demonstrated that NAC reduces extracellular cystine to cysteine, serving to support intracellular GSH biosynthesis and provide sulphydryl groups that enhance glutathione synthesis and glutathione-S-transferase activity [77,78]. Moreover, NAC could also directly scavenge hydroxyl radicals through thiol groups. However, thiol radicals could be generated by the interaction of NAC with reactive radicals, implying a pro-oxidant role for NAC. Generation of thiyl radicals can happen in several ways and is also influenced by different factors, such as oxygen, the presence of metals, and the origin of the sulfur component; therefore, the context could influence the antioxidant or pro-oxidant properties of NAC [79]. During CSE treatment, bone cells increase ROS production, and this oxidative stress context may induce NAC to protect cells from free radical damage. However, under physiological oxidative stress conditions, NAC can also have negative effects on bone cells due to pro-oxidative properties. Additionally, it is known that a lower amount of free radicals is necessary to induce bone progenitor cell osteogenic differentiation; therefore, reducing bone cells from free radicals in physiological conditions could negatively influence bone homeostasis [80].

Some limitations of this study need to be recognized. Bone healing is a dynamic process that is influenced by various factors, including other organs, vascular systems, and biochemical signals; our scaffold could not simulate all those biological conditions observed in vivo. Additionally, although SCP-1 and THP-1 cell lines stably proliferate and display the physiological function of bone cells, there are still some differences in gene expression and cell function between the cell line and primary cells. In our study, we explored the effects of MBE or GE in combination with CSE; however, the effect of the co-administration of MBE and GE on CSE-induced bone cell damage remains to be investigated. We observed that the therapeutic effects of antioxidants (MBE and GE) were more pronounced regarding functionality as opposed to viability. Similar results were achieved with NAC. Further investigations will focus on whether there are other mechanisms by which antioxidants reduce CSE-induced cell damage; for example, involving the BMP signaling pathway, Wnt/β-catenin signaling pathway, MAPK signaling pathway, and TGF-β signaling pathway. Although the CSE used in the study was an aqueous extract that better mimics the absorption of molecular species into the blood in smokers’ lungs and was sterile-filtered before application to the system such that particles with a size >0.2 µm were removed (comparable to the barrier function in the lung), our system could not represent the exposure of bone tissue in smokers. The addition of endothelial cells, immune cells, lung cells, or even liver cells would improve the model and better represent the molecular species or additional metabolites that are transported into the bloodstream and arrive at the bone tissue.

## 5. Conclusions

By activating the Nrf2 signaling pathway and reducing the RANKL: OPG ratio, MBE and GE can improve CSE-impaired osteogenesis and decrease CSE-induced osteoclastogenesis, thus preventing the disruption of bone homeostasis. MBE and GE mitigated CSE damage to bone cells to an extent comparable to NAC (precursor of glutathione). Therefore, MBE and GE show promise as potential bone health supplements for orthopedic patients who smoke.

## Figures and Tables

**Figure 1 antioxidants-11-02460-f001:**
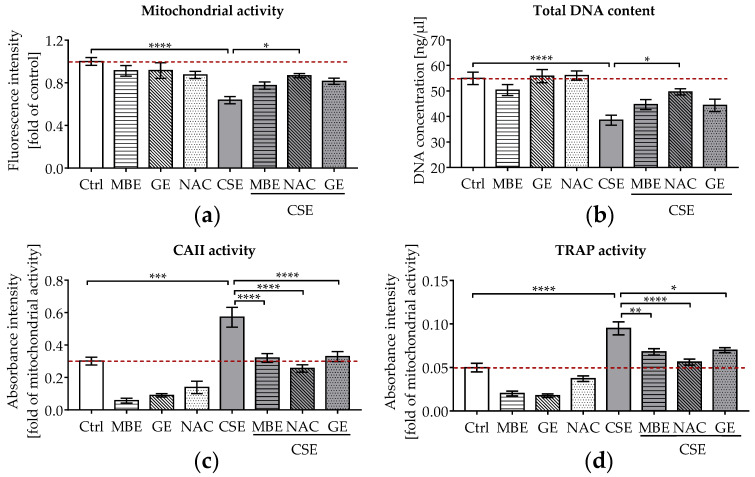
The effects of MBE and GE on SCP-1/THP-1 co-culture system exposed to CSE. The co-culture system was co-incubated with 5% CSE and (or) antioxidants (1.5 μg/mL MBE, 50 μg/mL GE, and 3.5 mM NAC) for 14 days. (**a**) Mitochondrial activity of the co-culture system was measured on day 14. (**b**) Total DNA content of the co-culture system was measured on day 14. For osteoclast function, CAII activity (**c**) and TRAP activity (**d**) were compared after 14 days of co-culture. Statistical differences were determined using two-way ANOVA test followed by Turkey’s multiple comparisons. Data are present as means ± SEM, and the significances are shown as * *p* < 0.05, ** *p* < 0.01, *** *p* < 0.001, and **** *p* < 0.0001 vs. CSE group. (*N* = 3–4, *n* = 3).

**Figure 2 antioxidants-11-02460-f002:**
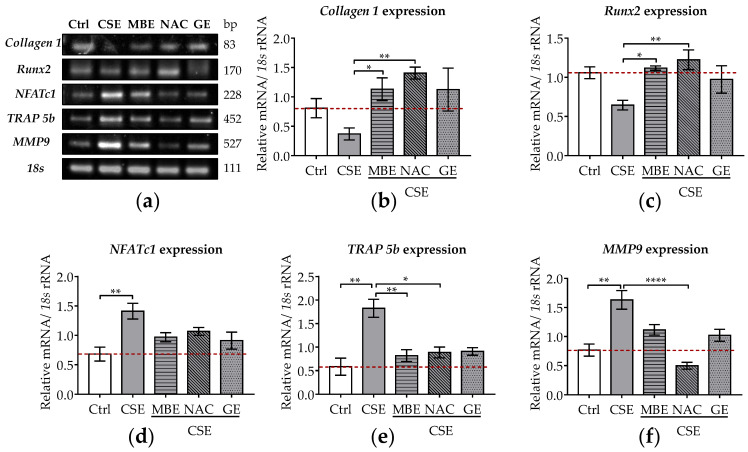
The expression levels of osteoblastic and osteoclastic genes in the co-culture system under exposure to CSE and antioxidants. SCP-1/THP-1 co-culture system was exposed to 5% CSE with or without antioxidants for 7 days. The gene expression levels were evaluated by RT-PCR. 18s rRNA served as a housekeeper gene. (**a**) Representative RT-PCR image showed the gene expression of *Collagen 1*, *Runx2*, *NFATc1*, *TRAP 5b*, and *MMP9* in 3D co-culture system with different treatments. Expression levels of *Collagen 1* (**b**), *Runx2* (**c**), *NFATc1* (**d**), *TRAP 5b* (**e**), and *MMP9* (**f**) mRNA were measured on day 7. Statistical differences were determined using the Kruskal–Wallis test followed by Dunn’s multiple comparison test. Data are presented as means ± SEM, and the significance is shown as * *p* < 0.05, ** *p <* 0.01, and **** *p* < 0.0001 vs. CSE group. *N* = 3, *n* = 2.

**Figure 3 antioxidants-11-02460-f003:**
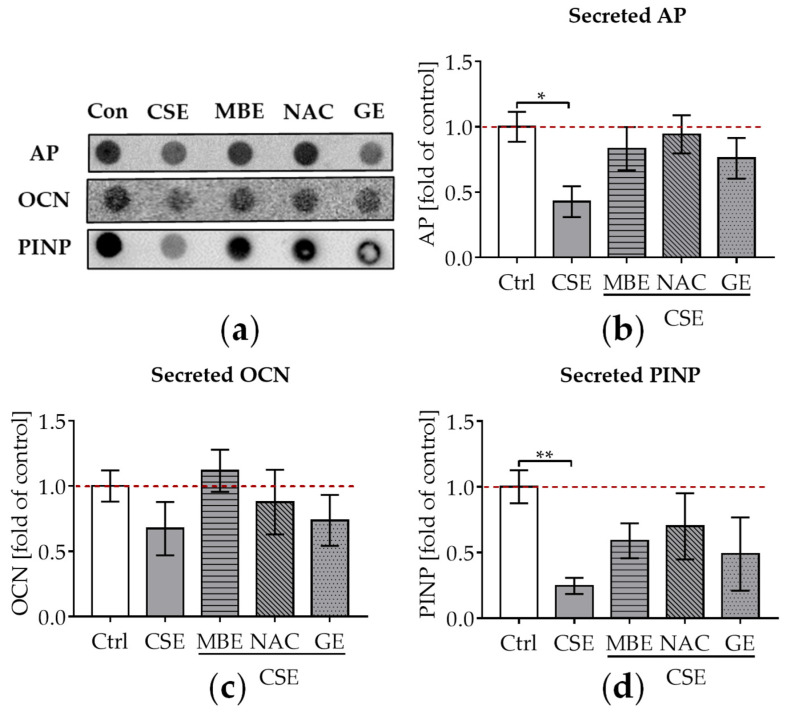
The influence of MBE and GE on CSE-impaired bone-forming function. After co-culture with osteogenic medium (2% FCS, 200 μM L-ascorbic acid 2-phosphate, 5 mM β-glycerolphosphate, 25 mM HEPES, 1.5 mM CaCl_2_, and 5 μM cholecalciferol) for 14 days, dot blot was performed to detect the secreted AP, OCN, and PINP in the supernatant. (**a**) A representative dot blot image showed the secreted protein levels of AP, OCN, and PINP in the 3D co-culture system with different treatments. (**b**) Secreted level of AP protein was measured in the co-culture system by day 14. (**c**) Secreted levels of OCN protein were measured in the co-culture system by day 14. (**d**) Secreted levels of PINP protein were measured in the co-culture system by day 14. Statistical differences were determined using the Kruskal–Wallis test followed by Dunn’s multiple comparison test. Data are presented as means ± SEM, and the significance is shown as * *p* < 0.05 and ** *p* < 0.01, vs. CSE group. *N* = 3, *n* = 3.

**Figure 4 antioxidants-11-02460-f004:**
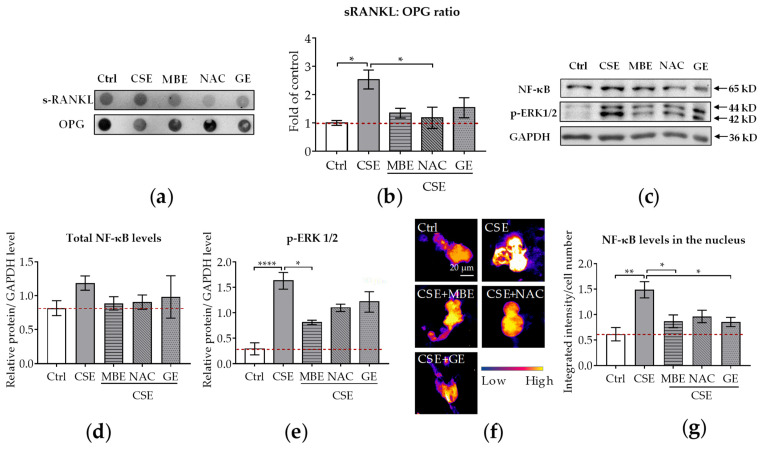
The sRANKL/OPG ratio and NF-κB expression in the co-culture system. Dot blot was used to detect the secreted sRANKL and OPG levels in the co-culture system by day 14 (*N* = 3, *n* = 3). Western blot (*N* = 3, *n* = 2) and immunofluorescence staining (*N* = 3, *n* = 3) were performed to detect the NF-κB protein level in both thee co-culture system and nucleus on day 7. (**a**) A representative dot blot image showing the secreted protein levels of sRANKL and OPG in the supernatant. (**b**) The ratio of sRANKL and OPG in the co-culture system. (**c**) A representative Western blot image showing the total protein levels of NF-κB and p-ERK1/2 in the co-culture system. (**d**,**e**) Quantitative analysis of total NF-κB and p-ERK1/2 protein levels in the co-culture system. (**f**) A representative immunofluorescence image showing the protein expressions of NF-κB (scale bar  =  20 μm). (**g**) The NF-κB protein level in the nucleus of the osteoclast. Statistical differences were determined using the Kruskal–Wallis test followed by Dunn’s multiple comparison test. Data are presented as means ± SEM, and the significance is shown as * *p* < 0.05, ** *p* < 0.01 and **** *p* < 0.0001 vs. CSE group.

**Figure 5 antioxidants-11-02460-f005:**
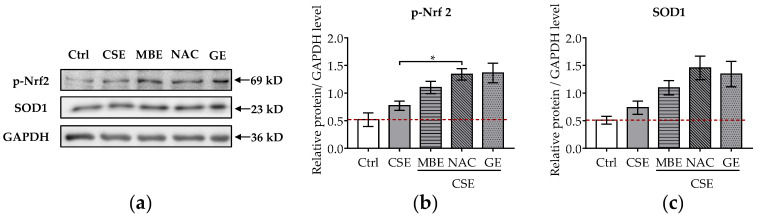
Influence of MBE and GE on antioxidant signaling pathway. Western blot was performed to detect the protein expression levels of p-Nrf2 and SOD1 in the co-culture system by day 7. (**a**) Representative Western blot image showing the protein levels of p-Nrf2 and SOD1 among different groups. The levels of p-Nrf2 (**b**) and SOD1 proteins (**c**) were measured in the co-culture system on day 7. Statistical differences were determined using the Kruskal–Wallis test followed by Dunn’s multiple comparison test. Data are presented as means ± SEM, and the significance is shown as * *p* < 0.05 vs. CSE group. *N* = 3, *n* = 2.

**Figure 6 antioxidants-11-02460-f006:**
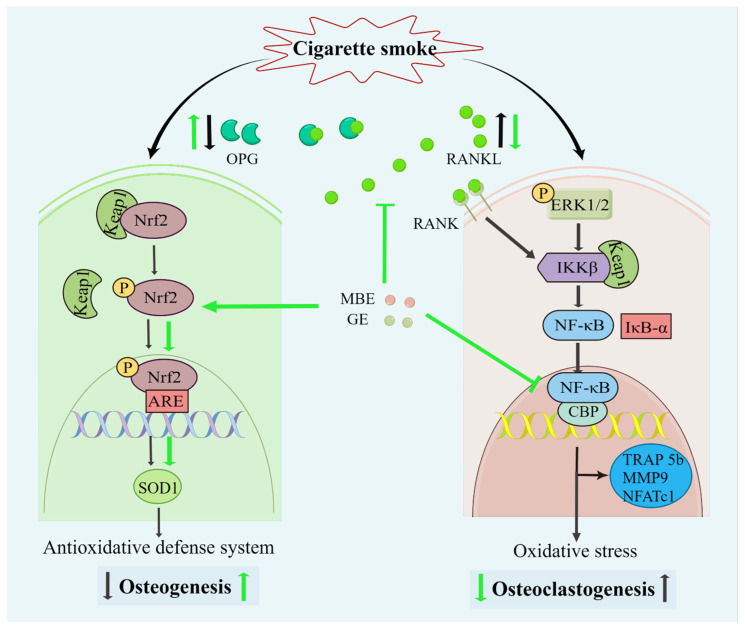
Diagram of the effects of MBE and GE on preventing CSE-induced bone cell damage. Both CSE and antioxidants could activate the Nrf2 signaling pathway. The dissociation of Keap1/Nrf 2 complexes prevents the degradation of transcription factor Nrf2 and, consequently, results in the activation of Nrf2. Combined with antioxidant response element (ARE) promoter, phosphorylated Nrf2 (p-Nrf2) triggers the antioxidative transcription of SOD1. Then, the antioxidative defense system enhances osteogenesis in the SCP-1/THP-1 co-culture system. However, the slight activation of the Nrf2 signaling pathway by CSE could not inhibit the oxidative damage in osteoblasts. This led to decreased OPG secretion by osteoblasts and, thus, an increase in the RANKL: OPG ratio. Through the interaction between RANKL and RANK (RANKL receptor), increased RANKL activated the NF-κB pathway. In addition, CSE also upregulated the NF-κB pathway in the co-culture system. CSE promotes the expression of phosphorylation ERK 1/2 via the production of reactive oxygen species (ROS). ROS-induced inhibitor of kappa B kinase beta (IKKβ) results in the degradation of IκB-α and the activation of NF-κB. Transcriptional factor NF-κB promoted the expression of osteoclast-related genes and osteoclastogenesis. The black and green arrows represent the activity of CSE and antioxidants (MBE or GE), respectively.

**Table 1 antioxidants-11-02460-t001:** Primer sequences and PCR conditions for target genes.

Gene	Accession Number	Forward Primer (5′–3′)	Reverse Primer (5′–3′)	Length (bp)	Ta (°C)	Cycles
*Collagen 1*	NM_000088.3	CAGCCGCTTCACCTACAGC	TTTTGTATTCAATCACTGTCTTGCC	83	56	35
*Runx2*	NM_001024630.4	CTGTGGTTACTGTCATGGCG	GGGAGGATTTGTGAAGACGGT	170	60	35
*NFATc1*	NM_172390.2	TGCAAGCCGAATTCTCTGGT	CTTTACGGCGACGTCGTTTC	228	64	40
*TRAP 5b*	NM_001111035.1	TTCCAGGAGACCTTTGAGGA	TAGGCAGTGACCCCGTATGT	452	58	35
*MMP9*	NM_004994.3	ATGAGCCTCTGGCAGCCCCT	CCGTGCTCCGCGACACCAAA	527	60	35
*18s rRNA*	NR_003286	GGACAGGATTGACAGATTGAT	AGTCTCGTTCGTTATCGGAAT	111	56	25

**Table 2 antioxidants-11-02460-t002:** The antibodies used for dot blot.

Antibody	Order#	Company	Dilution
Soluble receptor activator of nuclear factor kappa-B ligand (sRANKL)	sc-11383	Santa Cruz, Heidelberg, Germany	1:1000
Osteoprotegerin (OPG)	500-P149	Peprotech, Hamburg, Germany	1:1000
Procollagen type I N-terminal propeptide (PINP)	abx131414	Abbexa, Aachen, Germany	1:1000
Alkaline phosphatase (AP)	sc-23430	Santa Cruz, Heidelberg, Germany	1:1000
Osteocalcin (OCN)	sc-365797	Santa Cruz, Heidelberg, Germany	1:1000
Goat anti-rabbit IgG-HRP	sc-2004	Santa Cruz, Heidelberg, Germany	1:10,000
Donkey anti-goat IgG	sc-2020	Santa Cruz, Heidelberg, Germany	1:10,000
Goat anti-mouse IgM	sc-2064	Santa Cruz, Heidelberg, Germany	1:10,000

**Table 3 antioxidants-11-02460-t003:** The antibodies used for Western blot.

Antibody	Order#	Company	Dilution
Phospho nuclear factor erythroid-2-related factor-2 (p-Nrf2)	ab76026	Abcam, Cambridge, UK	1:1000
Nuclear factor kappa-B (NF-κB)	sc-109	Santa Cruz, Heidelberg, Germany	1:1000
Phospho extracellular regulated protein kinases1/2 (p-ERK1/2)	4370	Cell Signaling, Massachusetts, USA	1:1000
Superoxide dismutase 1 (SOD1)	sc-11407	Santa Cruz, Heidelberg, Germany	1:1000
Glyceraldehyde-3-phosphate dehydrogenase (GAPDH)	sc-365797	Santa Cruz, Heidelberg, Germany	1:1000
Goat anti-rabbit IgG-HRP	sc-2004	Santa Cruz, Heidelberg, Germany	1:10,000

## Data Availability

The data presented in this study is contained within the article and Appendix A.

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
