# Peer review of "Maqui Berry and Ginseng Extracts Reduce Cigarette Smoke-Induced Cell Injury in a 3D Bone Co-Culture Model"

_antioxidants, 2022, doi:10.3390/antiox11122460_

Round 1

Reviewer 1 Report

The paper presented is interesting well organized and written. I suggest to publish the paper in Antioxidants after addressing few comments.

-Materials and methods, 2.1 details on purchased chemicals should  be given (names, compounds, purity and so on), as well as details on cell culture mediums.

-no number of independent experiments was provided for the assays performe.

-Results

based on the MBE and Ge extract could be interesting understanding the percentage of active ingredients and if the beneficial effects could be ascribable to one/two compounds in order to give the possibility to open novel study including only such compounds. Maybe the indication of the vendors (see the point above) can be useful for speculating in this sense, if possible.

Reviewer 2 Report

Guo et al. employed a bone model established using gelatin/hydroxyapatite composite scaffold seeded with MSC and pre-osteoclast cell lines to examine the effects of maqui berry and ginseng extracts. The topic is relevant and interesting. However, the following concerns need to be addressed prior to further evaluation of its suitability for publication.

1.       Experiment design: a key feature of the bone model was the use of osteoblast- and osteoclast-like cells. How was the cell ratio determined? In he discussion and introduction, a clear explanation of OB-OC interactions and crosstalk would enhance the clarity and motivation of the study.

2.       Culture medium: the authors used osteogenic medium to co-culture MSCs and osteoclast precursors. This medium composition does not support osteoclast function. Please justify the selectin of medium and repeat the experiment with a more appropriate medium composition if needed.

3.       The osteogenic quality and osteoclast function were not clearly described. Histology would be necessary to show the deposition of calcium phosphate in the construct. Immunostaining is also highly desirable to stain the expression of relevant osteogenic and osteoclast marker proteins.

4.       Following my previous question, it is not clearly stated how the culture duration as determined. Was 7/14 days sufficient for the cells to fully differentiate and for the model to be physiological relevant?

5.       Line 392-395: how were these conclusions drawn from the figure?

Reviewer 3 Report

The manuscript entitled  “Maqui berry and ginseng extracts reduce cigarette smoke-induced cell injury in a 3D bone co-culture model” contains an extensive study about the point described in the title, added with the effect of N-AcetylCysteine (NAC), that is used as positive control. Studies on osteoblasts and osteoclast indicate the involvement of the Nrf 2 signaling pathway in the action of the antioxidant agents.

 Methods are appropriate and well described. Markers of osteoclast function and differentiation are correct. Osteoblastic genes (Collagen 1 and Runx2) are appropriate. The pertinent choices can be also applied to the markers of the transcription factors related to the oxidative response and antioxidant responses (NFkB and NRF2). Results are also clear, WB have good quality and the discussion is consistent with the obtained results.

Conclusion is brief and clear. MBE and GE might show promise as potential bone health supplements for orthopedic smoker patients. However, these results are obtained using a co-culture cell model, where the accessibility of the active (toxic and protective components) to the cells is easy. The point is how much of the active components of CSE do reach bone cells in vivo and the pharmacokinetics of those tobacco smoke components. Any data inserted at the discussion would be very helpful to assess the importance of the studies.

 Minor points before publication

Cytotoxicity tests of MBE, GE, and NAC

It is described that cell viability showed a remarkable reduction when the NAC concentration exceeded 7 mM, but is also reported that 3.5 mM NAC had a protective role in CSE-induced bone cell injury.  The change from protective to cytotoxicity seems to be in a narrow range (from 3.5 to 7 mM). The reasons of that would be briefly discussed in the light of the NAC mechanism of action. In turn, the protective action is not observed at Figure 1S (1e and 1f), so that the meaning of the word “protective” is misleading. Does that mean protection against CSE?

Figure 5: the meaning of the color (green and black) in the arrows of the scheme would be very helpful to understand the proposed mechanism of action of CSE on osteoblasts and osteoclasts.

Round 2

Reviewer 1 Report

my concerns were addressed in the revised version

Reviewer 2 Report

Thank you for taking the time to revise the manuscript. The authors have addressed all my concerns.

Reviewer 3 Report

The comments and requirements described in the review report have been fully addressed in the reply letter, and the manuscript has been amended according to these and other points required by my colleagues during the reviewing process.  The added material contributes to improve the quality of the study.